# The Chimeric Binjari-Zika Vaccine Provides Long-Term Protection against ZIKA Virus Challenge

**DOI:** 10.3390/vaccines10010085

**Published:** 2022-01-06

**Authors:** Jessamine E. Hazlewood, Bing Tang, Kexin Yan, Daniel J. Rawle, Jessica J. Harrison, Roy A. Hall, Jody Hobson-Peters, Andreas Suhrbier

**Affiliations:** 1Inflammation Biology Group, QIMR Berghofer Medical Research Institute, Brisbane, QLD 4029, Australia; Jessamine.Hazlewood@qimrberghofer.edu.au (J.E.H.); Bing.Tang@qimrberghofer.edu.au (B.T.); Kexin.Yan@qimrberghofer.edu.au (K.Y.); Daniel.Rawle@qimrberghofer.edu.au (D.J.R.); 2School of Chemistry and Molecular Biosciences, University of Queensland, St Lucia, QLD 4072, Australia; j.harrison1@uq.edu.au (J.J.H.); roy.hall@uq.edu.au (R.A.H.); 3Australian Infectious Disease Research Centre, GVN Center of Excellence, The University of Queensland and QIMR Berghofer Medical Research Institute, St Lucia, QLD 4067, Australia

**Keywords:** vaccine, Binjari virus, Zika virus, mouse model, insect-specific flavivirus, chimeric virus

## Abstract

We recently developed a chimeric flavivirus vaccine technology based on the novel insect-specific Binjari virus (BinJV) and used this to generate a chimeric ZIKV vaccine (BinJ/ZIKA-prME) that protected IFNAR^-/-^ dams and fetuses from infection. Herein, we show that a single vaccination of IFNAR^-/-^ mice with unadjuvanted BinJ/ZIKA-prME generated neutralizing antibody responses that were retained for 14 months. At 15 months post vaccination, mice were also completely protected against detectable viremia and substantial body weight loss after challenge with ZIKV_PRVABC59_. BinJ/ZIKA-prME vaccination thus provided long-term protective immunity without the need for adjuvant or replication of the vaccine in the vaccine recipient, both attractive features for a ZIKV vaccine.

## 1. Introduction

Zika virus (ZIKV) is a mosquito-transmitted flavivirus and the etiological agent of congenital Zika syndrome (CZS), a constellation of primarily neurological birth defects that includes microcephaly [1,2]. During 2015–2017, over 0.5 million Zika cases are estimated to have occurred in the Americas, leading to ≈4000 cases of confirmed CZS [3]. CZS is associated with considerable social and economic burdens for both affected children and their caretakers [4,5]. A large range of activities are thus underway internationally to develop ZIKV vaccines, with an emphasis on safety, given that the target population for vaccination will include pregnant women and/or women of child-bearing age [6,7,8,9].

We recently developed a chimeric flavivirus vaccine technology based on the insect-specific flavivirus, Binjari virus (BinJV). BinJV is a lineage II insect-specific flavivirus isolated from *Aedes normanensis* mosquitos in northern Australia in 2013 [10]. BinJV emerged to be remarkably tolerant of substitution of its pre-membrane and envelope (*prME*) genes with the *prME* genes of flaviviruses that cause disease in humans [11]. The chimeric viruses are able to replicate efficiently in insect cells (allowing production in C6/36 cells), but are unable to replicate in vertebrate cells. For mammalian vaccine recipients, they might thus be viewed as a replication-defective virus-like-particle vaccine, which comprises the prME proteins of the target flavivirus, the capsid protein of BinJV, and the RNA of the chimeric virus [11,12]. Flavivirus vaccines utilizing the BinJV chimera technology have been shown to induce protective immunity in a number of pre-clinical models of infection and disease, including West Nile virus [13], dengue virus [14], ZIKV [12,15], and yellow fever virus [16]. Flaviviruses replicate poorly in wild-type mice, requiring the aforementioned protection studies to be conducted in mice defective for type I interferon (IFN) responses such as in type I IFN receptor knockout (IFNAR^-/-)^ mice. Herein, in a new experiment, we extend our previous short-term immunity and protection studies [12,15] to illustrate that a single vaccination with the chimeric BinJV—ZIKV vaccine (BinJ/ZIKA-prME) provides long-term neutralizing antibody responses and protection against infection and diseases in female IFNAR^-/-^ mice.

## 2. Materials and Methods

### 2.1. Ethics Statement

Mouse work was undertaken in accordance with the Australian Code for Care and Use of Animals for Scientific Purposes, as outlined by the National Health and Medical Research Council of Australia. Animal work was approved by the QIMR Berghofer Medical Research Institute Animal Ethics Committee (Approval: A1604-611M).

### 2.2. The Vaccine and Vaccination

The chimeric BinJ/ZIKA-prME viral cDNA was generated using a modified Circular Polymerase Extension Reaction (CPER) protocol and was transfected into C6/36 cells [12]. The BinJ/ZIKA-prME virus was recovered and amplified in C6/36 cells, with virus particles purified using a sucrose cushion and a potassium tartrate gradient to generate the vaccine preparation as described [15]. The BinJ/ZIKA-prME construct contained the *prME* genes of ZIKV_Natal_ [17] (encoding amino acid Ala123 to Ala794; Genbank ID: KU527068) with the rest of the genome of the chimera derived from BinJV (Genbank ID: MG587038). The structure of the BinJ/ZIKA-prME virus particles closely resembles the structure of ZIKV particles [12].

IFNAR^-/-^ mice on a C57BL/6J background [18], bred in-house at QIMR Berghofer MRI, were anesthetized and vaccinated i.m. into both quadriceps muscles (50 µL per muscle), with 10 µg of BinJ/ZIKA-prME or BinJ/ZIKA-prME formulated with adjuvant. Control mice received 50 µL of PBS or 50 µL PBS with adjuvant. The adjuvant was generated in-house to resemble AS01, with each mouse receiving 2 µg of 3-O-desacyl-4′-monophosphoryl lipid A (Sigma-Aldridge, St Louis, MO, USA) and the saponin QS-21 (Creative Biolabs, Shirley, NY, USA) for prime and boost, as described [16]. 

### 2.3. Neutralizing Antibody Response Determination

Serum neutralization assays (using ZIKV_PRVABC59_) were undertaken by incubating dilutions of heat-inactivated mouse serum (in duplicate, 50 µL/well of a 96-well plate) with 100 CCID_50_ of virus (50 µL/well) for 2 h before the addition of Vero E6 cells (Sigma-Aldridge, ECACC Vero C1008) (10^4^ in 100 µL/well). The culture medium used throughout was RPMI 1640 (Thermo Fisher Scientific, Scoresby, VIC, Australia) supplemented with 2% fetal calf serum (Sigma-Aldrich). After a 5-day incubation at 37 °C and 5% CO_2_, plates were fixed, cytopathic effects were assessed by crystal violet staining, and 50% neutralizing titers were determined as described [16].

### 2.4. ZIKV Challenge of IFNAR^-/-^ Mice

Mice were challenged subcutaneously with 10^4^ CCID_50_ of ZIKV_PRVABC59_ (GenBank ID: KU501215) [15] (kindly provided by Dr S. Tajima (Department of Virology I, National Institute of Infectious Diseases, Tokyo, Japan [19]) at 15 months post vaccination. The challenge virus was produced in C6/36 cells, with the culture medium (as above) determined to be endotoxin-free by RAW264-HIV-LTR-LUC Bioassay [20]. Virus stocks were checked for mycoplasma as described [21] and cells were checked for mycoplasma using MycoAlert™ (Lonza, Basel, Switzerland). Mice were weighed and bled for viremia determinations using CCID_50_ assays, as described [15]. Mice were euthanized with CO_2_ when weight loss reached or approached 20%, an ethically defined endpoint.

### 2.5. Statistical Analysis

IBM SPSS Statistics (IBM Corp., Armonk, NY, USA) for Windows, Version 22.0 was used for statistical analyses. The non-parametric Kolmogorov–Smirnov test was used for viremia and neutralization titer data. For weight change over time, the repeat measures ANOVA was used. For survival, the Kaplan–Meier log-rank test was used.

## 3. Results

### 3.1. Maintenance of Neutralizing Antibody Responses after BinJ/ZIKA-prME Vaccination

Female IFNAR^-/-^ mice were vaccinated i.m., with BinJ/ZIKA-prME, BinJ/ZIKA-prME formulated with adjuvant, or control vaccinations, PBS or PBS formulated with adjuvant, and were then bled at the indicated times for assessment of neutralizing antibody responses (Figure 1a).

BinJ/ZIKA-prME vaccination induced significant and persistent ZIKV-specific neutralizing antibody responses, with titers (for BinJ/ZIKA-prME with no adjuvant) above the level of detection (1 in 30 serum dilution) for all mice and time points tested (Figure 1b). However, by 14 months, titers had waned significantly (Figure 1b, *p* = 0.031). Formulation with adjuvant provided no significant benefit (Figure 1b). Results were normalized for assay-to-assay variability using a positive reference serum that was included in each neutralization assay run. No neutralizing antibody responses were detected in the control PBS-vaccinated mice (Figure 1b). We have previously shown that mice vaccinated with BinJV as a negative control also showed no detectable induction of ZIKV-specific neutralizing antibody responses [15], with ZIKV and BinJV prME proteins showing only a 42% amino acid identity.

### 3.2. BinJ/ZIKA-prME Vaccination Provides Long-Term Protection against Infection and Disease

At 15 months post vaccination mice were challenged with ZIKV_PRVABC59_. None of the BinJ/ZIKA-prME vaccinated mice (with or without adjuvant) showed any detectable viremia on any of the days tested, with both control groups showing clearly detectable viremias (Figure 1c). Viremias in these older mice were actually comparable to those seen in younger mice [15], an observation consistent with a recent study showing no significant correlations between mouse age and viremia in this model [22]. Although two of the mice receiving BinJ/ZIKA-prME + Adjuvant had neutralizing antibody titers at 14 months that were below the level of detection (Figure 1a), they were nevertheless entirely protected against detectable viremia (Figure 1c), presumably because low titers of neutralizing antibodies [23] and/or ZIKV prME-specific CD4 T cells [24] are able to provide protection. 

The control PBS vaccinated mice showed a substantial and significant drop in body weight after challenge, whereas the weight loss in BinJ/ZIKA-prME vaccinated mice (with or without adjuvant) was minimal (Figure 1d). A body weight drop reaching 20% for any individual animal represents an ethically defined endpoint requiring euthanasia. None of the BinJ/ZIKA-prME vaccinated mice required euthanasia, whereas 75–80% of control mice were euthanized (Figure 1e). Although reaching a 20% reduction in body weight, the mice that were euthanized showed no other overt signs of serious illness.

## 4. Discussion

We illustrate herein that a single vaccination of IFNAR^-/-^ mice with unadjuvanted BinJ/ZIKA-prME was able to induce protective immune responses that lasted for 14/15 months. Live-attenuated ZIKV vaccines have previously been shown to provide long-term protective immunity [25,26]. However, herein, we show that BinJ/ZIKA-prME was able to generate such immunity, with our previous studies showing that BinJ/ZIKA-prME has no detectable capacity to replicate in mammalian cells [12]. We saw no evidence for replication of BinJV or BinJV chimeras in vertebrate cells including IFNAR^-/-^ MEFs. We also saw no evidence of replication or persistence of BinJ/ZIKA-prME in IFNAR^-/-^ mice, or in B, T, and NK deficient mice [12]. Another insect-specific flavivirus (Aripo virus) was recently shown to induce robust innate immune responses in wild-type macrophages, dominated by type I IFN responses, despite also being unable to replicate in vertebrate cells [27]. Type I IFN responses can provide potent adjuvant activity [28,29] that can promote robust and long-lasting protective immunity [30,31]. However, type I IFN responses would likely be blunted in the IFNAR^-/-^ mice used herein as the type I IFN receptor-dependent feed-forward amplification and dissemination of type I IFN responses [32] would be absent. Nevertheless, many genes ordinarily induced by type I IFNs can also be induced in their absence [33,34,35]. Whether these are induced after BinJ/ZIKA-prME vaccination sufficiently to generate the long-term immunity described herein remains unclear. The virus-like-particle nature of the BinJ/ZIKA-prME vaccine may also contribute, with such vaccines recently reported to promote long-term plasma cell responses [36].

Why adjuvant provided no benefit for the BinJ/ZIKA-prME vaccine, whereas the same adjuvant did provide significant benefit for antibody induction in IFNAR^-/-^ mice after vaccination with the Binjari-yellow fever virus vaccine (BinJ/YF-prME) [16], remains unclear. Stability and maintenance of an authentic structure of flavivirus-like-particles may represent a key issue for efficient induction of neutralizing antibody responses by these kinds of vaccines [37]. Conceivably, the BinJ/ZIKA-prME viral particles are particularly stable [12,38] and the BinJ/YF-prME vaccine is perhaps less stable. A different adjuvant also provided only marginal and setting-dependent benefit for BinJ/WNV_KUN_-prME vaccination in CD1 mice [13], with cryo-electron microscopy arguing that this chimera was stable and structurally authentic [39]. Alternatively or in addition, different levels of immunopotentiating contaminants may be co-purified with these different chimeric vaccines, which may provide varying levels of adjuvant activity.

A limitation of many flavivirus mouse challenge models, such as the one used herein, is that they require the use of mice defective for type I IFN responses, such as IFNAR^-/-^ mice [15,16] or AG129 mice [14], which are also defective for type II IFN responses. Unfortunately, there is no consensus on how vaccine-induced immune responses are impacted by the lack of the type I IFN receptor in IFNAR^-/-^ mice [40]. However, one might speculate that, if anything, responses to RNA-containing vaccines might be dampened in IFNAR^-/-^ mice given the aforementioned loss of the feed-forward amplification of type I IFN responses in these animals.

In summary, a single BinJ/ZIKA-prME vaccination provided long-term protection without the use of adjuvant or replication of BinJ/ZIKA-prME in the vaccine recipient, both attractive features for a ZIKV vaccine [6,9].

## Figures and Tables

**Figure 1 vaccines-10-00085-f001:**
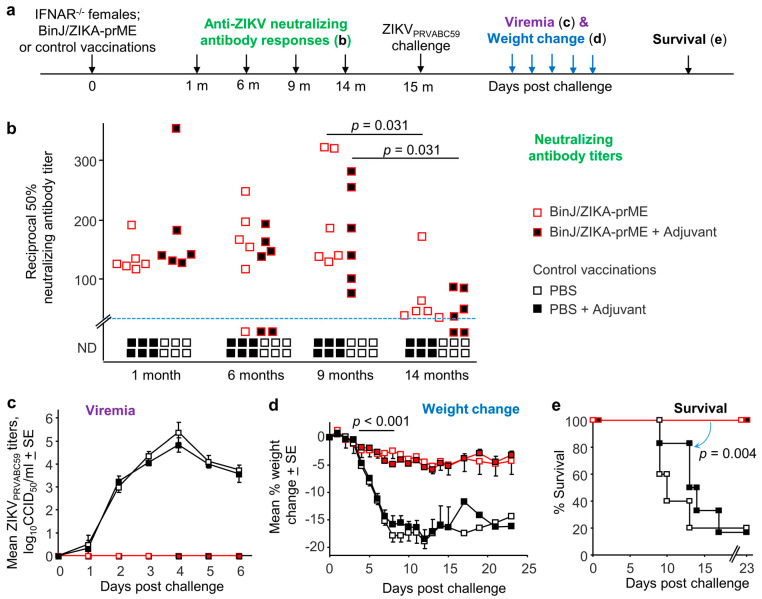
Vaccination, antibody assays, and ZIKV challenge. (**a**) Schematic illustration of the timeline of the study. (**b**) Anti-ZIKV_PRVABC59_ 50% neutralizing antibody titers for individual mice. ND—Not Detected. Limit of detection was a 1 in 30 dilution of serum. Adjuvant provided no statistically significant improvement at any time point. BinJ/ZIKA-prME vs. Controls *p* = 0.031 (6 months), *p* = 0.005 (1, 9 and 14 months). Nine vs. 14 months, *p* = 0.031 for both vaccine groups. Statistics by Kolmogorov–Smirnov tests. (**c**) Mean viremias after s.c. challenge with ZIKV_PRVABC59_ (n = 6 mice per group, except PBS n = 5). The limit of detection was 2 log_10_CCID_50_/_mL_ per mouse. BinJ/ZIKA-prME vaccinated mice vs. Control vaccinated mice, *p* = 0.005 for days 2 to 5; statistics by Kolmogorov–Smirnov tests. (**d**) Mean percent body weight changes; with percent change for each mouse calculated relative to day 0 for that mouse. For BinJ/ZIKA-prME vaccinated mice vs. Control mice; repeat measures ANOVA days 4 to 8, *p* < 0.001. (**e**) Survival. Mice were euthanized when they reached ethically defined end points. Statistics by Kaplan–Meier log-rank tests. All the data were derived from one group of mice.

## Data Availability

Data is contained within the article.

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
