# Peer review of "The Chimeric Binjari-Zika Vaccine Provides Long-Term Protection against ZIKA Virus Challenge"

_vaccines, 2022, doi:10.3390/vaccines10010085_

Round 1
Reviewer 1 Report
Hazlewood and Tang et al. are following up on the publication “A recombinant platform for flavivirus vaccines and diagnostics using chimeras of a new insect-specific virus” from 2019 in Science Translational Medicine. In their submitted manuscript the authors describe that the chimeric Zika virus vaccines based on Binjari virus provides long-term protection (14/15 months) against Zika virus infection in mice. Overall the manuscript is interesting, but it seems in order to fully understand the manuscript one has to look up many other published papers and given information is not always complete. The introduction is missing the importance of the study. Also, I am not completely convinced that the submitted manuscript fits to the topic "Animal Model in Biomedical Research" as the animal model used in this study is not discussed much in the proposed work.
Introduction:
The introduction is missing the point why a vaccine against Zika virus is important. What kind of virus is it? How is it transmitted? Case numbers? Lethality? Why would a Zika vaccine be important? Why is there no Zika vaccine? What is Binjari virus? I understand that the authors have published similar work before, however, readers might not have read their other works nor are experts on flaviviruses and will need more information to be interested and follow the topic. I highly recommend to give more information in the introduction and state the importance of it.
Methods:
How exactly does the BinJ/Zika-prME look like? Are these particles or subviral particles? Can they replicate? What Zika strain was cloned into this vaccine? Which amino acids # are considered to be prME according to the authors? A graphic would be helpful. The neutralization assay is one of the most important tests in this paper and the methods is not explained at all in the manuscript. What cell lines were used? How was that Zika virus prepared? What are the GenBank numbers of the viruses?
Results:
Figure 1B and C – how many replicate of the tests were done? Is this the mean? Are the difference with and without adjuvant statistically significant?
As some work was published before by the authors regarding this I wonder if these are the same mice from a previous study? If it is a follow up from a previous study, the authors should note that in the manuscript.
Discussion:
Line 109 to 112 – Here the discussion relates to data which is not shown in this work but in the previous one. I feel the wording should be more clear that these information comes from a previous study. This could be e.g. part of the introduction as it is not really related to the new data in this work. It would be useful to know in the beginning that BinJ virus is not replicating in mammalian cells. How well does this vaccine design approach compare to many other Zika virus vaccines approaches in the long-term? Is it doing better? Why would it be better or worse?
Author Response
- I am not completely convinced that the submitted manuscript fits to the topic "Animal Model in Biomedical Research" as the animal model used in this study is not discussed much in the proposed work.
We have added a section in the Introduction and in the Discussion regarding the use of IFNAR mice models, implications and potential shortcomings.
-
The introduction is missing the point why a vaccine against Zika virus is important. What kind of virus is it? How is it transmitted? Case numbers? Lethality? Why would a Zika vaccine be important? Why is there no Zika vaccine? What is Binjari virus? I understand that the authors have published similar work before, however, readers might not have read their other works nor are experts on flaviviruses and will need more information to be interested and follow the topic. I highly recommend to give more information in the introduction and state the importance of it.
We have now included all this information in the Introduction with some recent references in the field.
-
How exactly does the BinJ/Zika-prME look like? Are these particles or subviral particles? Can they replicate? What Zika strain was cloned into this vaccine? Which amino acids # are considered to be prME according to the authors? A graphic would be helpful.
These details have been added to Introduction and Methods. A graphic is presented in the STM paper and the previous Vaccines paper - not sure we need to provide this again.
- The neutralization assay is one of the most important tests in this paper and the methods is not explained at all in the manuscript. What cell lines were used? How was that Zika virus prepared? What are the GenBank numbers of the viruses?
Full details are now provided in a new section in the Methods
- Figure 1B and C – how many replicate of the tests were done? Is this the mean? Are the difference with and without adjuvant statistically significant?
This long term study was conducted once only, with one group of mice. Fig. 1b shows individual mouse data, Fig. 1c, d show means; this has been clarified in the figure legends and y axes.
L 127 states “Formulation with adjuvant provided no significant benefit”.
- As some work was published before by the authors regarding this I wonder if these are the same mice from a previous study? If it is a follow up from a previous study, the authors should note that in the manuscript.
Now clarified in the last paragraph of the Introduction
- Line 109 to 112 – Here the discussion relates to data which is not shown in this work but in the previous one. I feel the wording should be more clear that these information comes from a previous study.
Rephrased to make clear what is from this and previous studies.
- This could be e.g. part of the introduction as it is not really related to the new data in this work. It would be useful to know in the beginning that BinJ virus is not replicating in mammalian cells.
This information has now been provided in the Introduction
- How well does this vaccine design approach compare to many other Zika virus vaccines approaches in the long-term? Is it doing better? Why would it be better or worse?
A really good question but unfortunately there is no suitable available data on which to base a meaningful comparison. Long term studies such as the one presented are not frequently performed, with side by side comparisons also rarely countenanced by commercial entities.
Reviewer 2 Report
The manuscript is a follow-up from the results by Hobson-Peters et al published in Sci Transl Med in 2019. In this instance, the authors explore the potential benefit of using an adjuvant consisting of 3-O-desacyl-4’-monophosphoryl lipid 54 A and the saponin QS-21.
In the parent publication by Hobson-Peters as well as in this one, the flavivirus vaccines are never compared to the empty vector (i.e., BinJV). Instead they use PBS. I think PBS is an inadequate negative control for virus-vectored vaccines. In this regard,
- How can the authors assess the specific contribution of immune responses to ZIKV proteins in protecting mice in these experiments?
- The authors comment on the possible role of CD4 T cells in providing protection, Can they comment or provide data on the CD4 T cell-cross reactivity of BinJV and ZIKV (if any)?
Secondly, if the adjuvant isn't working as the authors hoped or thought it would be. What are the significantly novel additions of this publication, that are not already defined in the Hobson-Peters publication?
Author Response
- The manuscript is a follow-up from the results by Hobson-Peters et al published in Sci Transl Med in 2019. In this instance, the authors explore the potential benefit of using an adjuvant consisting of 3-O-desacyl-4’-monophosphoryl lipid 54 A and the saponin QS-21. In the parent publication by Hobson-Peters as well as in this one, the flavivirus vaccines are never compared to the empty vector (i.e., BinJV). Instead they use PBS. I think PBS is an inadequate negative control for virus-vectored vaccines. In this regard, How can the authors assess the specific contribution of immune responses to ZIKV proteins in protecting mice in these experiments?
We have previously shown that mice vaccinated with BinJV as a negative control also showed no detectable induction of ZIKV-specific neutralizing antibody responses (Hazlewood, 2020) with ZIKV and BinJV prME proteins showing only a 42% amino acid identity. We have added this information to the Results.
- The authors comment on the possible role of CD4 T cells in providing protection, Can they comment or provide data on the CD4 T cell-cross reactivity of BinJV and ZIKV (if any)?
BinJV vaccinated mice were also not protected against ZIKV challenge (Hazlewood, 2020). Although we have not formally done an assessment of T cell responses, the high level of divergence (only 42% sequence identity in prME proteins) between BinJV and ZIKV (with no long stretches of homology) would make extensive cross-reactivity unlikely.
The BinJ/ZIKA-prME vaccine contains only BinJV capsid protein. In the vaccine recipient no detectable BinJV proteins are made after BinJ/ZIKA-prME vaccination; no detectable RNA replication means minimal templates for translation. So the CD4 responses we mean here are ZIKV prME-specific – we have clarified this in the text.
- Secondly, if the adjuvant isn't working as the authors hoped or thought it would be. What are the significantly novel additions of this publication, that are not already defined in the Hobson-Peters publication?
Adjuvant is arguably supposed to improve long-term antibody responses – clearly for BinJ/ZIKA-prME this did not manifest. What is new in this publication is the surprising longevity of the protective neutralising antibody response after a single vaccination with this vaccine without adjuvant.
Reviewer 3 Report
The manuscript by Hazlewood et al is a nice continuation on their previous work on the chimeric BinJ/ZIKA-prME vaccine published in Vaccines (Basel) last year (ref 9) . Reading their previous work is very helpful in understanding this paper. Most of the technical details are not included. For example. the age of the mice when the get vaccinated (I guess 2-4 months from previous paper). For the neutralisation assay method, the authors cite the previous paper ref 9, which then cite another previous manuscript from this group, which is actually reference 6 in this manuscript. It could save the reader time if reference 6 is cited directly. However that is not an open access article and therefore some reader won’t be able to access it. As this is a brief article, I would recommend to add more technical details, and avoid this inconvenience.
The type of analysis done is similar to their previous work, however it could be interesting to check whether they can detect ZIKV RNA in tissues (brain, reproductive organs) as well as in blood at the termination point. Obviously this is only a suggestion if tissue were collected and store at the time of the experiment.
Another observation worth of a comment is the age of the mice when they are infected, which should be around 20 months(?). For most murine spp this is almost old age. Is this a valid model? This is a genuine question, I have no idea of the answer, but maybe worth mentioning in the discussion.
It is interesting and useful that protective and long last immunity can be generated with a single vaccination. Have the authors in previous work looked into vector induced immunity? Can this platform being used more than once? Again, a sentence in the introduction will be helpful if this has already been addressed.
Author Response
- The manuscript by Hazlewood et al is a nice continuation on their previous work on the chimeric BinJ/ZIKA-prME vaccine published in Vaccines (Basel) last year (ref 9). Reading their previous work is very helpful in understanding this paper. Most of the technical details are not included. For example. the age of the mice when the get vaccinated (I guess 2-4 months from previous paper). For the neutralisation assay method, the authors cite the previous paper ref 9, which then cite another previous manuscript from this group, which is actually reference 6 in this manuscript. It could save the reader time if reference 6 is cited directly. However that is not an open access article and therefore some reader won’t be able to access it. As this is a brief article, I would recommend to add more technical details, and avoid this inconvenience.
This was also mentioned by the other reviewers – we have added a much higher level of detail into the Methods, including a new section for the neutralization assays
- The type of analysis done is similar to their previous work, however it could be interesting to check whether they can detect ZIKV RNA in tissues (brain, reproductive organs) as well as in blood at the termination point. Obviously this is only a suggestion if tissue were collected and store at the time of the experiment.
An important point. This experiment has already been reported (Hobson-Peters, 2019 ) and we saw no evidence of persistence in IFNAR or other immune deficient mice. We have added this observation to the Discussion.
- Another observation worth of a comment is the age of the mice when they are infected, which should be around 20 months(?). For most murine spp this is almost old age. Is this a valid model? This is a genuine question, I have no idea of the answer, but maybe worth mentioning in the discussion.
Yes we should not have stated “the expected viremia” as there is no expectation that viremia in these much older mice would be comparable with viremia in much younger mice. We have rephrased, noting that viremias are actually comparable to those seen in younger mice, with a recent paper also showing no significant correlation between of age and viremia in this IFNAR mouse model. Age is perhaps more of an issue in wild-type mouse models than in IFNAR mice.
- It is interesting and useful that protective and long last immunity can be generated with a single vaccination. Have the authors in previous work looked into vector induced immunity? Can this platform being used more than once? Again, a sentence in the introduction will be helpful if this has already been addressed.
The Introduction now explains that this technology essentially resembles virus-like-particle technology comprising target prME immunogens assembled into a virus-like-particle, but containing chimeric viral RNA. Thus there is no vector as such, with only BinJV capsid present in the vaccine, with no detectable transcription from the chimeric viral RNA in vertebrate cells.
Round 2
Reviewer 1 Report
Thanks to the author for responding to my points.
Minor point:
Line 84 to 91 - please check the grammar.
Author Response
We have rephrased the indicated paragraph to improve clarity. We have also had the manuscript read by a professional proof reader who has made some additional minor suggestions to improve clarity.